# Task Sharing and Remote Delivery of Brief Interpersonal Counseling for Venezuelan Migrants and Refugees Living in Peru during the COVID-19 Pandemic: A Mixed-Methods Pilot Study

**DOI:** 10.3390/ijerph21020166

**Published:** 2024-01-31

**Authors:** M. Claire Greene, Mayra Muro, Jeremy C. Kane, Erin Young, Alejandra Paniagua-Avila, Lucy Miller-Suchet, Maria Nouel, Annie G. Bonz, Maria Cristobal, Matthew Schojan, Peter Ventevogel, Bryan Cheng, Silvia S. Martins, Jose Carlos Ponce de Leon, Helen Verdeli

**Affiliations:** 1Mailman School of Public Health, Columbia University, New York, NY 10032, USA; 2HIAS Peru, Lima 15034, Peru; mayra.muro.j@gmail.com (M.M.);; 3Teachers College, Columbia University, New York, NY 10026, USA; 4HIAS, Silver Spring, MD 20910, USAmatthew.schojan@hias.org (M.S.); 5United Nations High Commissioner for Refugees, 1201 Geneva, Switzerland

**Keywords:** mental health and psychosocial support, interpersonal counseling, migrants and refugees

## Abstract

Refugees and migrants experience an elevated risk for mental health problems and face significant barriers to receiving services. Interpersonal counseling (IPC-3) is a three-session intervention that can be delivered by non-specialists to provide psychological support and facilitate referrals for individuals in need of specialized care. We piloted IPC-3 delivered remotely by eight Venezuelan refugee and migrant women living in Peru. These counselors provided IPC-3 to Venezuelan refugee and migrant clients in Peru (n = 32) who reported psychological distress. Clients completed assessments of mental health symptoms at baseline and one-month post-intervention. A subset of clients (n = 15) and providers (n = 8) completed post-implementation qualitative interviews. Results showed that IPC-3 filled a gap in the system of mental health care for refugees and migrants in Peru. Some adaptations were made to IPC-3 to promote its relevance to the population and context. Non-specialist providers developed the skills and confidence to provide IPC-3 competently. Clients displayed large reductions in symptoms of depression (d = 1.1), anxiety (d = 1.4), post-traumatic stress (d = 1.0), and functional impairment (d = 0.8). Remote delivery of IPC-3 by non-specialists appears to be a feasible, acceptable, and appropriate strategy to address gaps and improve efficiency within the mental health system and warrants testing in a fully powered effectiveness study.

## 1. Introduction

In recent years, the trifecta of political instability, economic collapse, and social unrest has sparked a severe humanitarian crisis in Venezuela, leading to one of the largest mass migrations in the history of the region [1]. Within the past decade, over seven million people are estimated to have fled Venezuela to other countries, with over six million having remained within Latin America [2]. During the COVID-19 pandemic, Peru was the recipient of the second highest influx of Venezuelan migrants, refugees, and asylum seekers [3,4]. At the same time, Peru was also one of the countries most affected by COVID-19, having the highest case fatality rates, incidence and number of infections reported in Latin America [5,6,7]. 

Refugees, migrants, and asylum seekers are at increased risk of mental health and psychosocial problems [8,9,10]. Many Venezuelans have experienced potentially traumatic events, stress, and anxiety due to the adversity they have faced in their home country [11]. The stress of leaving behind family and friends, adapting to a new culture, and finding work and housing in a new country while often encountering discrimination and xenophobia can also take a toll on mental health [12]. The COVID-19 pandemic has further exacerbated mental health challenges for refugees, asylum seekers, migrants, and the host community in Peru [13,14].

Mental health services, provided by the government in Peru, non-governmental organizations (NGOs) and international agencies, are limited [15,16]. Their availability and access were further restricted during the COVID-19 pandemic due to mitigation measures (such as social distancing) and limited resources [17,18]. Task sharing is a capacity-building strategy of mental health care and psychosocial support delivery by community members within their own communities that has been widely adopted to overcome mental health specialist shortages, barriers to help seeking, and approachability of services [19]. There is growing evidence supporting the feasibility, effectiveness, and quality of implementation (e.g., provider competency, intervention fidelity) of mental health and psychosocial interventions delivered through a task sharing model [20,21,22,23]. While the majority of studies employing a task sharing model have implemented an in-person model of intervention delivery, there has been an emerging body of research exploring digitally assisted task sharing models [24]. The COVID-19 pandemic, together with the increasingly protracted and mobile nature of forced displacement, has prompted the adoption of digital solutions to address barriers to in person mental health interventions with various populations including refugees [25,26,27,28,29].

Interpersonal counseling (IPC-3), a brief three-session version of Interpersonal Psychotherapy, is a scalable psychological intervention for symptoms of common mental health problems that can be delivered through a task sharing model [30,31,32]. IPC-3 is used as an entry point (i.e., the first step) within a stepped care system that targets distress for people with mild mental health problems while also providing initial support and engagement in individuals who may benefit from a more intensive mental health intervention [31]. IPC-3 focuses on addressing interpersonal problems that trigger or exacerbate psychological distress, categorized into four categories (problem areas): grief, interpersonal disputes/disagreements, role transitions, and social isolation/loneliness [33]. There is robust evidence supporting the effectiveness of IPT in reducing depressive symptoms in humanitarian contexts [34,35,36,37,38,39]. Fewer studies have evaluated IPC [31], yet previous research among displaced women in Colombia suggest that brief versions of IPT (such as IPC-3) are feasible and acceptable as a first level intervention within a stepped-care system [40,41]. The objective of this pilot study was to adapt IPC-3 for Venezuelan migrants in Peru and evaluate its preliminary effectiveness and implementation when delivered remotely by non-specialist providers.

## 2. Materials and Methods

### 2.1. Study Setting

Venezuelans form the largest group of forcibly displaced persons in Peru. As of 2021, it is estimated that Peru hosts over 1.3 million Venezuelans in need of international protection, with the majority living in urban areas such as Lima, Arequipa, and Trujillo [42]. In recent years, the Peruvian government has taken steps to support the Venezuelan community. For example, in 2019, Peru launched the “Temporary Permanence Permit”, which allows Venezuelans to stay and work legally in the country for up to two years. Peruvian law also guarantees the right to mental health care for all individuals, including migrants and refugees. The Ministry of Health has developed a National Mental Health Plan [43], which includes a focus on improving mental health services for vulnerable populations, including migrants and refugees. In 2019, the government launched the Mental Health in Emergencies and Disasters program which provides psychosocial support to refugees and migrants, including individual and group counseling, as well as support for community-based mental health initiatives.

Overall, while there are policies in place to ensure that displaced populations in Peru have access to mental health services, there are still significant challenges related to accessing these services. As a result of these challenges, many Venezuelan refugees and migrants in Peru remain in need of humanitarian assistance, including access to basic services and support for mental health and psychosocial needs. NGOs such as HIAS Peru and UN agencies such as UNHCR are working to provide support to the Venezuelan community in Peru, with a particular focus on women, children, and survivors of violence. HIAS is a global humanitarian organization that supports refugees and other displaced persons around the world. In Peru, HIAS focuses on providing a range of services to refugees and asylum seekers, including legal assistance, psychosocial support, and other forms of social assistance. One key area of focus for HIAS Peru is mental health and psychosocial support (MHPSS) programming, including individual and group counseling, and other forms of emotional support. HIAS Peru also works to build the capacity of local organizations to provide MHPSS services to refugees and other vulnerable populations. This includes providing training to mental health professionals and working to raise awareness about the importance of MHPSS among government officials, policymakers, and the general public.

### 2.2. Participants and Procedures

#### 2.2.1. Pre-Implementation Adaptation and Implementation Planning

The first phase of the study included semi-structured qualitative interviews with key stakeholders to explore the appropriateness, acceptability, suggested adaptations, and implementation considerations for IPC-3 to address the mental health and psychosocial needs of Venezuelan refugees and migrants in Peru. We conducted 10 key informant interviews with Venezuelan refugees and migrants living in Peru, program managers, and local health providers/counselors. Eligible individuals were 18+ years of age and known to HIAS through engagement in previous MHPSS programming as knowledgeable about the mental health and psychosocial problems affecting migrants, refugees, and asylum seekers in their community. We excluded individuals unable to provide informed consent. HIAS MHPSS staff contacted potential participants to inform them about the study. If participants were interested, HIAS MHPSS staff referred them to research staff for informed consent and enrollment. Once enrolled, participants completed an in-depth interview remotely with a member of the research team (MM, JCP). Interviews were recorded and then transcribed using NVivo following the end of the interview.

#### 2.2.2. Training and Supervision

The second phase of the study involved selecting and training community members as IPC-3 providers. Providers were adult (18+ years) Venezuelan refugees and migrants with a high school education (or equivalent) who were living in Peru and were willing to participate in IPC-3 training and provide IPC-3 to members of their community. We excluded individuals who planned to relocate in the foreseeable future. HIAS MHPSS staff identified eligible community members from the refugees and migrants who have engaged with their programming and through their network of community-based organizations. Selected and interested providers completed an initial three-day remote training led by psychologists at Columbia’s Teachers College (HV, EY, BC). All providers completed knowledge tests individually at the end of training and were required to achieve minimum knowledge levels (75% correct responses) before proceeding with implementation and supervision. Providers were allocated into two groups, each of which participated in weekly two-hour remote group supervision led by psychologists at Teachers College. Providers were paid a stipend for their time in training, supervision, and implementation.

#### 2.2.3. Implementation of IPC-3

All providers were required to provide IPC-3 to at least three cases as part of their training and supervision. IPC-3 clients were identified by HIAS psychologists during routine client screenings. Eligible clients were 18+ years of age, residing in Peru, and reported elevated psychological distress. Elevated psychological distress was operationalized as moderate anxiety (Generalized Anxiety Disorder-7, GAD-7 > 10) [44] and/or depressive symptoms (Patient Health Questionnaire-9, PHQ-9 > 10 and <20) [45,46]. Participants were excluded if they displayed cognitive impairment, severe psychiatric symptoms that would not be appropriate for IPC-3, reported high risk of suicide (Columbia Suicide Severity Rating Scale, CSSRS >= 3), or were unable to provide consent. HIAS psychologists provided eligible participants with information about the study and, if interested, referred them to a member of the research team. Eligible and enrolled participants were allocated to IPC-3 providers with the goal of maintaining equivalent caseloads across providers. Each participant completed an intake assessment (week 1), three weekly sessions of IPC-3 (weeks 2–4), and a 1-month follow-up assessment (week 8) with their assigned IPC-3 provider. All assessments and IPC-3 sessions were conducted remotely.

#### 2.2.4. Post-Implementation

All IPC-3 providers (n = 8) and a subset of IPC-3 clients (n = 15) completed post-implementation semi-structured interviews to explore their perceptions of IPC-3 and its impacts, as well as recommendations for further adaptation and implementation. A subset of 15 clients was purposively selected using maximum variation sampling to reflect different levels of engagement and response to IPC-3. Selected clients and all providers completed an in-depth interview with a member of the HIAS research team (MM, JCP). All interviews were recorded and then transcribed using NVivo.

A summary of the study procedures is provided in Figure 1.

### 2.3. Measures

Qualitative, semi-structured interview guides were designed using the Johns Hopkins Applied Mental Health Research Group Dissemination and Implementation measure and adapted to the language and culture in Peru [47]. Additional questions were added to explore mental health needs and existing organizations. Interview guides covered the following implementation outcomes: appropriateness/relevance, acceptability, feasibility, implementation (e.g., remote delivery, provider characteristics, barriers/facilitators), as well as recommended adaptations. Post-implementation interview guides also included questions exploring scalability and sustainability of IPC-3.

The FRAME Adaptation Tool was used to record adaptations made throughout the study period by members of the research team during weekly project management meetings [48,49].

IPC-3 providers administered the following assessments of mental health symptoms and functional impairment to IPC-3 clients at the intake and follow-up assessment: Patient Health Questionnaire 9 (PHQ-9) to assess depressive symptoms [45,46], Generalized Anxiety Disorder 7 (GAD-7) to assess anxiety symptoms [44], the PTSD Checklist Civilian Version (PCL-C) to assess post-traumatic stress symptoms [50], and the World Health Organization Disability Assessment Schedule (WHODAS) to assess functional impairment [51]. The Columbia Suicide Severity Rating Scale (CSSRS) [52] was administered if participants reported a score of two or greater on the ninth item of the PHQ-9, which assesses suicidal ideation. The PHQ-9 and GAD-7 were used as the screening tools to assess eligibility. In addition to the intake and follow-up assessments, the PHQ-9 was administered at the beginning of every IPC-3 session to monitor symptoms throughout the intervention.

All measures were administered in Spanish. Table 1 summarizes the measures used to evaluate each of the study outcomes according to the RE-AIM framework.

### 2.4. Analysis

Adaptations recorded using the FRAME tool were described using pre-defined codes across the following domains: phase of implementation, whether the adaptation was proactive or reactive, who was involved in the decision to make the adaptation, the level of delivery affected by the adaptation, whether the adaptation was fidelity-consistent, the nature of the modification, the goal/reason for the adaptation, and any contextual factors that influenced the adaptation [48,49]. Codes were applied in real-time when the adaptations were recorded throughout study implementation. We performed qualitative thematic analysis of interview transcripts with key informants, providers, and clients. Three research members with expertise in psychology (MM, JCP), primary care (APA), and implementation research (APA) developed a codebook. We started by conducting individual open coding to identify main ideas that were revised and discussed iteratively among the team to identify and agree on emerging themes. Emerging themes were related to context, RE-AIM framework dimensions [54], and overarching adaptations, barriers and facilitators to IPC-3 implementation. We piloted and refined the codebook and performed line-by-line coding with NVivo Software (Version 1.7.1). Finally, we developed qualitative memos summarizing the range of responses by type of participant for each qualitative theme (MM, APA, LMS).

Quantitative process indicators (e.g., attendance, attrition) were calculated from study monitoring forms. The distribution of demographic characteristics and mental health of IPC-3 clients at baseline were presented as descriptive statistics (Mean and standard deviation for continuous variables, number of participants and proportion for categorical variables) using data from the intake assessment. We estimated the change in mental health outcomes over time using non-parametric Wilcoxon Signed Rank Tests comparing baseline to follow-up changes. We integrated qualitative and quantitative results on IPC-3 client outcomes using a converged mixed-methods analysis approach and generated a joint display combining the estimated and reported outcomes among clients over time.

### 2.5. Ethics

All participants provided written informed consent prior to enrollment. All procedures reviewed and approved by the Institutional Review Boards at the Columbia University Medical Center, Teachers College, and Prisma (Peru Ethics Committee).

## 3. Results

### 3.1. Description of the Sample

This study enrolled 10 key informants who completed pre-implementation interviews, nine IPC-3 providers who completed post-implementation interviews, and 32 IPC-3 clients who completed quantitative assessments of their mental health throughout the study period and post-implementation interviews (Table 2). Key informants were 37.8 years of age, on average (SD = 6.1), all were female, most were from Venezuela (70.0%) and had been in Peru for an average of 4.7 years (SD = 0.7), most had a college degree (80.0%), and most worked and/or consulted for non-governmental organizations (70.0%), UN organizations (20.0%), or were community leaders (10.0%). Similarly, IPC-3 providers were 36.4 years of age, on average (SD = 7.6), all were female and from Venezuela, and they had been in Peru for an average of 2.9 years (SD = 0.8). About half had a college degree (55.6%) and most worked for non-governmental organizations (33.3%), were community leaders/volunteers (22.2%), or both (44.4%). The IPC-3 clients were 36.5 years of age, on average (SD = 12.2, Range: 23–71), and most were female (96.9%). All clients were from Venezuela and had been living in Peru for 2.9 years, on average (SD = 1.4). Almost half were unemployed (43.8%) and most of those who were employed had informal jobs (37.5%) followed by part-time (12.5%) and full-time (6.2%) positions. Approximately 19% of clients had previously received mental health and psychosocial support services, whereas most key informants (90.0%) and providers (77.8%) had reported prior utilization of mental health and psychosocial support services.

### 3.2. Reach and Accessibility of Mental Health Services among Migrants and Refugees in Peru

The main system-level barrier to accessing MHPSS identified by study participants was the limited number of organizations and providers offering mental health services to Venezuelan refugees and migrants in Peru. There were few NGOs with psychologists and staff offering mental health services to Venezuelan refugees and migrants. Those that did mainly provided basic support and crisis intervention sessions primarily for adults, with fewer options available for children and adolescents. For specialized care, refugees and migrants must be referred to state-run health facilities, which were often overcrowded and difficult to access. IPC-3 clients who had previously attempted to access mental health services reported additional barriers such as long wait times for appointments, long distances to health facilities, a lack of connection with providers, and direct and indirect costs (e.g., transportation fares) of mental health services. Some Venezuelan refugees and migrants in Peru had specialized training and clinical experience in psychotherapy and, upon arrival to Peru, continued to provide services in a paid or volunteer role. These providers actively participated in community organizations serving the Venezuelan migrant communities to address these system-level barriers to access.

### 3.3. Appropriateness, Acceptability, and Anticipated Reach of IPC-3 for Migrants and Refugees and Peru

During pre-implementation interviews, key informants described several mental health needs of the Venezuelan population in Peru that are not being adequately addressed by existing services including mental stress, depression, and migratory grief. Migratory grief was considered the root of emotional instability and other mental health problems that was exacerbated by the challenges faced in the host country as well as limited support for integration. Key informants believed that IPC-3 would be appropriate for these needs. This was later confirmed by providers in post-implementation interviews who frequently reported that role transitions were the primary problem area that their clients struggled with. Key informants also acknowledged the potential limits of IPC-3 in being able to address more serious problems, including serious depression, suicidal thoughts, psychosis, or people experiencing violence.


*“I think that these four [IPC-3 strategies] fit very well with refugees and migrants. There will always be grief, not grief but as they call it migratory grief [duelo migratorio], role transitions, also the issue of isolation... All my patients are in transition now. It is very common, I believe that it does fit.”*
Key informant, Venezuelan migrant and psychosocial coordinator at an NGO, pre-implementation.

IPC-3 was also seen as a first step to engage with mental health services, and to serve as a triage for people in need of additional support. Key informants thought that IPC-3 would be particularly relevant for those who may not feel comfortable or be able to immediately access formal mental health services. Key informants noted that a brief intervention like IPC-3 delivered within communities could reach more people and have a large community impact.


*“Sometimes the NGOs take a long time to respond to help these [people] and we don’t know what to do. But [IPC-3] gives us more security on how to give this support to people while more help is coming, it gives us more security that what we are doing is really good for the person, it really helps them. In an easy way, in a reliable way we can calm and help the person, and maybe, who knows, this help will allow them to continue their activities and their live or even allow them to improve their mental health, their wellbeing.”*
Provider, and NGO worker/community leader, post-implementation.

### 3.4. Adoption of IPC-3 by Community-Based, Non-Specialist Providers in Peru

Prior to implementation key informants identified several characteristics that were important to consider when selecting IPC-3 providers. They recommended that IPC-3 providers be empathetic, good listeners, charismatic, committed, motivated, responsible, able to follow the steps and structure involved in IPC-3, and have a leadership position within their community. Most key informants were supportive of training Venezuelan refugees and migrants without prior experience in mental health interventions as IPC-3 providers. They reported that Venezuelan refugees and migrants who share a similar background, way of speaking, culture, and lived experience of migration would be a strength, which clients and providers confirmed in post-implementation interviews.


*“We identify with each other. Our experiences of how we have moved forward serve as support and a foundation for others to move forward. Seeing, as a migrant, other women, other people, men, who got ahead and who are using and supporting others with very effective tools, that helps us, it motivates us. In other words, if you get ahead, I can too. So there is a very positive identity.”*
Key informant, Venezuelan migrant and NGO Leader, pre-implementation.


*“[The IPC-3 provider] was very kind, sweet, understanding. And the fact is that she too, as a Venezuelan, has gone through the same or similar things to me. That gave me a lot of confidence to tell her my things and feel understood, even though I did not know her and had never seen her. And you see yourself in these people, these people are just like you.”*
IPC-3 client, post-implementation.

Key informants also noted that it could be challenging for the providers who may identify strongly with the difficult situations experienced by their clients. Key informants recommended that the trainers help the providers “know their limits” and how to refer participants if they needed additional support. For these reasons key informants recommend that providers be in a relatively stable socioeconomic situation at present.

Using the information provided by key informants, we invited ten providers to be trained as IPC-3 providers. Of those invited, nine completed the training. Seven providers passed the knowledge test the first time they took it and the remaining two passed the knowledge test the second time it was administered. The average passing score was 92%. One of the nine providers dropped out after completing two of the three training cases due to personal/family issues. In post-implementation interviews, providers noted how IPC-3 helped them personally, including helping them heal and move forward by applying the IPC-3 tools within their own lives and enriching their role within the community. They also appreciated the opportunity to improve their skills sets and the possibilities for their careers, including pursuing careers and further education in psychology. Providers also noted some of their concerns during the training and implementation of IPC-3. They spoke about the fear they experienced when they began their roles as a provider, but how with the support of the supervisors and project team, they realized they were capable of implementing IPC-3 well and helping participants.


*“I started to wonder if I was really going to make it, to be able to calm the person down. So the fact that you [project team] believed in me for the program moved me a lot. I didn’t expect it and the experience helped me a lot. It helped me to have more confidence in myself, a confidence that I had lost.”*
Provider and Community Leader, post-implementation.

Providers were motivated by the connections they made with their clients, the gratitude they received from their clients at the end of IPC-3, the improvements they observed in their clients’ lives, and the skills they gained through the process. They struggled when clients disengaged or did not complete IPC-3. Yet, through supervision, they realized that this does not reflect their capabilities as a provider, and there are many factors that influence a client’s decision to stop participating. Providers also noted several sacrifices they needed to make to fulfill this role including balancing their schedules and investing a significant amount of time, often during lunchbreaks or in the evening, to complete all the sessions.


*“As a [community] leader, you receive many people who need help, and sometimes you don’t know what to do because they are crying, desperate, distressed. Then you are left short, because you can only provide information to help them and sometimes you can give them some donations, but that’s all, the anguish is still there… I did not feel I had the strength or tools to help them, to give them support. So when you presented me with the project, what motivated me was that I understood that the IPC could help me develop those capacities to provide that help.”*
Provider and NGO worker/community leader, post-implementation.

### 3.5. Implementation of IPC-3 through a Task-Sharing, Remote Delivery Model

We made three adaptations to IPC-3 during the pre-implementation phase based on the recommendations of key informants (Table 3). First, we specified that Venezuelan migrants and refugees who were involved in community programs would be selected as providers to promote reach, appropriateness, and sustainability of IPC-3. Second, due to the COVID-19 context, we delivered IPC-3 remotely to improve reach, retention, feasibility, and to comply with social distancing guidelines. Lastly, we separated the screening process from IPC-3 intake and intervention sessions. Instead of having IPC-3 providers lead the recruitment and screening process, HIAS psychologists screened and referred eligible clients to IPC-3 providers. IPC-3 providers then completed the intake process, IPC-3 sessions, and follow-up assessments. This adaptation was made to align with the implementing organization’s service structure, policies and procedures.

Additional adaptations to the implementation of IPC-3 were made by the IPC-3 trainers and providers during training and supervision. These included adaptations to the IPC-3 manual and materials to ensure that the terminology maintained conceptual equivalence and contextual relevance. IPC-3 trainers supplemented routine training activities to support the providers in managing difficult situations (e.g., suicidality, confidentiality issues, provider–client boundaries) and to adjust how IPC-3 information was presented to better align with the sociocultural context. The study team also augmented the compensation for providers to account for the additional time required for training and supervision due to these adaptations. Lastly, during supervision, IPC-3 supervisors incorporated activities to monitor provider burden and provided additional support to improve provider satisfaction and acceptability. During post-implementation interviews, providers described some of the strategies and adaptations they made on a case-by-case basis including sending reminder messages to clients, giving clients homework following each session to keep them engaged, and providing additional information and resources to clients.

Despite these adaptations made during the pre-implementation and implementation phases of the project, some barriers to implementation remained. There were mixed perceptions regarding the dose and duration of IPC-3. Many key informants expressed concerns about the short duration of the intervention and that each of the three sessions were quite long (i.e., 90 min), which made them question the feasibility of maintaining engagement and impacts. Most providers noted that the initial sessions were often longer, but over time as the providers became more comfortable with the content and as they build rapport with the client, the duration of the sessions shortened. In general, providers had to be flexible with scheduling the sessions—including the time and frequency—to accommodate the clients’ schedules. Another barrier to participation for the clients was lack of financial resources. Some participants were unable to participate in sessions because they were facing financial pressures to support their family and often needed to utilize the time they had available to work or look for jobs.

Another unique aspect of IPC-3 implementation was the remote delivery. Most providers used phone calls or a mix of phone and video calls to provide IPC-3. The option to use video calls was often based on the stability of the internet connection for the client and the provider. Some providers noted other reasons for relying only on video calls including the participants needing to take the sessions outside of their homes, not wanting to show their home environment, and challenges downloading or using video call applications. Some clients did not have private spaces for the sessions so were often distracted during sessions.


*“Seeing them in the sessions, they are cooking with the TV on, with the child crying, with the dog barking. It is not a nice space to work in.”*
Provider, NGO worker, post-implementation.

Some clients did not have a personal cell phone, which made it difficult for the provider to reliably communicate with them. Multiple providers noted that when they lost connection during video calls it was very difficult to continue with the conversation, which also made phone calls often the preferred mode of delivery. While there were advantages to conducting the training and IPC-3 sessions remotely, many providers noted that they would have liked some opportunities for in-person activities, especially to build community amongst the providers themselves.

### 3.6. Preliminary Indicators of Effectiveness of IPC-3 on Mental Health Outcomes

We assessed 48 individuals for eligibility. Of those individuals, 32 were enrolled in IPC-3 and the remaining were ineligible (n = 15) or required referral to other services (n = 1). Two thirds of the 32 participants enrolled in IPC-3 (n = 21) completed all three sessions. Of the rest of the participants, five completed two sessions, three completed one session, and three did not attend any session. The one-month follow-up assessment to evaluate change in mental health outcomes over time was completed by 20 participants (Figure 2).

As shown in Figure 3 and Table 4, we observed significant reductions in depressive symptoms (d = 1.1; *p* < 0.001), anxiety symptoms (d = 1.4; *p* < 0.001), post-traumatic stress symptoms (d = 1.0, *p* < 0.001), and functional impairment (d = 0.8; *p* = 0.002) from baseline to the follow-up assessment. Depressive symptoms, which were measured at every session, displayed a linear reduction over the course of the three sessions.

These findings were corroborated by IPC-3 providers and clients during qualitative interviews (Table 4). Additional changes not captured by the quantitative measures included improvements in the adoption of health habits (e.g., healthy eating and sleeping patterns, absence of physical pain, enhanced self-efficacy and confidence, reduced anger or reactivity to stressful situations, improved problem-solving and coping skills, and better personal appearance and hygiene). Clients reported that these changes had also been noticed by other individuals in their lives, such as family members, and that many of these changes had persisted after the conclusion of the intervention.


*“I did not relate to other people. For me, it was very difficult to leave my house and now I leave my house, I go out, I share with other people. Maybe it is not that I go out every day to a party or to a neighbor’s house for a coffee, but yes I share with other people and I am not afraid to leave my house.”*
IPC-3 client, post-implementation.


*“To think that my family was there, in Venezuela, was very painful. When I was asked about it or remembered it, all I did was cry and cry. That is something that I have been able to overcome. The truth is that now I can talk about these issues in a calm way. I still feel sadness, but I can talk about these issues. I no longer cry inconsolably. The truth is that [IPC-3] helped me enormously, especially because in those sessions I could talk about subjects that I did not talk about with anyone because I would get into anguish and strong crying, a very big pain in my chest. But thanks to these sessions, I was able to do it.”*
IPC-3 client, post-implementation.

### 3.7. Maintenance of Community-Based Delivery of IPC-3

Key informants, providers, and clients emphasized the role of IPC-3 as one part of a stepped care system to promote scalability and sustainability, maintain quality and safety, and reduce barriers to engagement in mental health care. IPC-3 clients recognized the number of people in their community who could benefit from a program like IPC-3, but face many barriers to seeking and receiving mental health services.


*“There are many people in the community, Venezuelans, who really need this help, to be listened to, to have a space with someone they trust to talk about their things, their problems, or to talk about anything, but to talk. Most of us don’t talk, we just do what we have to do, take care of our children, go to work, and we don’t have time to open our hearts and tell what is happening to us.”*
IPC-3 client, post-implementation.

Providers described various resources that would be required to continue implementing IPC-3. First, providers indicated the need for a telephone (separate from their personal phone) or computer, internet access, physical space to work, and payment. Second, providers indicated that it was important that an organization with strong leadership provide support and structure to IPC-3 implementation. Being connected to an organization could improve referrals across levels of mental healthcare. Third, providers acknowledged the importance of having ongoing supervision from mental health professionals. Fourth, providers highlighted the essential informal support they received from their personal networks and from other providers, which was essential to maintaining their own wellbeing and ability to continue to deliver IPC-3.

## 4. Discussion

### 4.1. Summary of Findings

In this study we evaluated the implementation of IPC-3 delivered remotely by non-specialist providers to Venezuelan refugees and migrants in Peru who reported elevated levels of psychological distress. IPC-3 was appropriate to the mental health and psychosocial needs of Venezuelan refugees and migrants, acceptable within the cultural context, and feasible to implement using a remote delivery and task sharing approach. Although this study was not designed to definitively test efficacy and other implementation outcomes, we observed promising improvements in clinical and implementation outcomes that should be evaluated in future research.

### 4.2. Implications of Study Findings Related to the Feasibility, Acceptability, and Relevance of Remote Delivery of IPC-3 through Task Sharing

A brief, community-based intervention like IPC-3 appeared to fill a critical gap in MHPSS for Venezuelan refugees and migrants in Peru. The use of providers from the migrant community was consistently seen as a strength of the program. Several clients and providers recognized that the ability to identify with each other strengthened the trust and rapport, thus facilitating engagement and participation. The benefits of training community members as providers and delivering MHPSS in community-based settings have also been observed in other MHPSS studies conducted among refugees and migrants and displaced populations in Latin America [55,56,57]. Community-based MHPSS interventions delivered through task sharing have been tested in multiple refugee settings and has been recommended as a strategy to bridge gaps in the availability, appropriateness, and accessibility of MHPSS in these communities [19]. However, it is important to acknowledge the challenges with training community members as MHPSS providers. In this study, participants noted the psychological difficulties that providers may face when supporting members of their own community, many of whom have had very similar experiences to those of the providers. Additionally, we selected providers who were already in community leadership roles and thus they often had many competing priorities and responsibilities. These challenges underscore the importance of supportive supervision to ensure that providers needs are also met, and the program does not produce unintended harm among the providers themselves.

The remote modality facilitated continuity of care and overcame a range of barriers to participation that are particularly salient for migrant populations (e.g., mobility, inconsistent work schedules, etc.). However, some providers and clients noted that more in-person engagement would have enhanced their experience. A review of digital mental health interventions for refugees and immigrants found that most studies that have used remote delivery strategies have been conducted in high-income countries [25]. Therefore, the current study is one of the few to use employ this approach in a middle-income country. Similar to our findings, other studies have reported that digital mental health approaches are acceptable by refugee and migrant populations, can improve flexibility, save time, and are culturally sensitive. These studies have also found reductions in symptoms of depression and post-traumatic stress disorder [25]. However, researchers also caution against the rapid scale-up and scale-out of digital mental health interventions in the absence of an inclusive process of developing these tools with refugees and the careful consideration of the cultural context, assumptions, and potential pitfalls of the technology [58,59].

Our results support the relevance of IPC-3 for the mental health and psychosocial needs of refugees and migrants. Specifically, the symptoms and psychosocial problems reported during pre-implementation interviews aligned with the IPC-3 intervention targets and problem areas. For example, role transitions emerged as the most frequent problem area faced by IPC-3 clients in this study due to the cascades of life changes they experienced during migration and as they adjusted to their life in Peru. Furthermore, stress and depression, which key informants reported as some of the most prevalent mental health problems facing refugees and migrants in Peru, are common intervention targets for IPC-3 and we observed significant reductions in these symptoms over the course of the intervention. One common psychosocial problem expressed by refugees and migrants was ‘duelo migratorio’ [migratory grief], which presented differently from grief as it is described in IPC-3. This required some adaptations to the manual and additional training of the providers to maintain conceptual equivalence and fidelity to the intervention concepts.

### 4.3. Strengths and Limitations

This study is one of the first to test IPC-3 for displaced and migrant populations in Latin America. A previous study of IPC-3 as part of a stepped care model was conducted with internally displaced women in Colombia. Similar to our findings, the study in Colombia also found significant reductions in symptoms of depressive, anxiety, and post-traumatic stress symptoms [40,41]. However, an important limitation of these studies of IPC-3 in Colombia and the current study in Peru is that both were designed as non-controlled studies. Studies of IPC among non-migrant populations in Latin America have similarly shown reductions in mental health symptoms, but these changes are comparable to those observed among participants receiving enhanced usual care [60]. Other studies of longer versions of IPC conducted outside of Latin America and largely in clinical settings with populations experiencing other health conditions have revealed mixed results regarding the comparative effectiveness of IPC-3 to other interventions and usual care [31].

This pilot study possessed several limitations that must be considered when evaluating the results and their implications. First, the lack of a comparison condition and small sample size preclude the generalizability of study findings and limits our ability to attribute observed changes in study outcomes to IPC-3. While the study selected measures that have previously been used in Latin America and/or humanitarian contexts, they have yet to be validated within Venezuelan refugees and migrants in Peru. Further research addressing the key limitations of this study is needed to determine the effectiveness of the remote-delivered IPC-3 by non-specialists to migrant and refugee populations in Latin America. Despite these limitations, this study provides evidence of feasibility and can serve as a model for implementation in larger-scale studies.

## 5. Conclusions

Findings from this study support the feasibility of task sharing and remote service delivery models to increase access to MHPSS for hard-to-reach and vulnerable populations, including migrants and refugees, in community settings. We identified promising trends in clinical and implementation outcomes, which need to be evaluated rigorously using a fully powered, controlled study design to determine whether IPC-3 is effective in reducing symptoms of common mental disorders in this population and/or successfully link persons who need higher levels of care to treatment. The successful implementation of IPC-3 intervention and research procedures suggest that a definitive, evaluation of IPC-3 as part of a stepped care system is feasible and may serve as a model for other brief MHPSS interventions in diverse communities.

## Figures and Tables

**Figure 1 ijerph-21-00166-f001:**
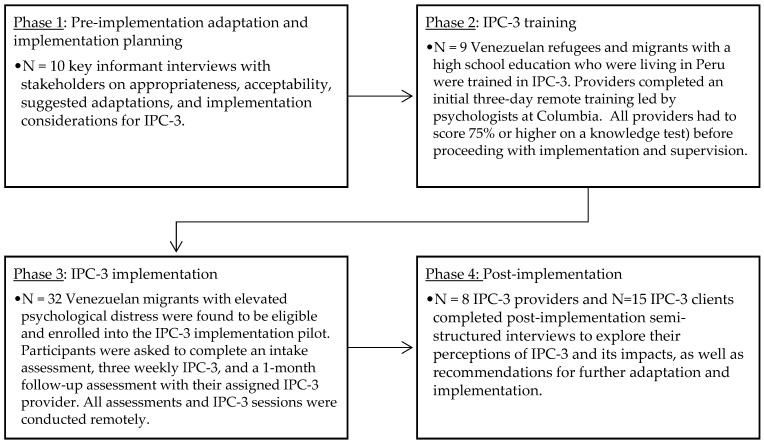
Summary of study phases and methods.

**Figure 2 ijerph-21-00166-f002:**
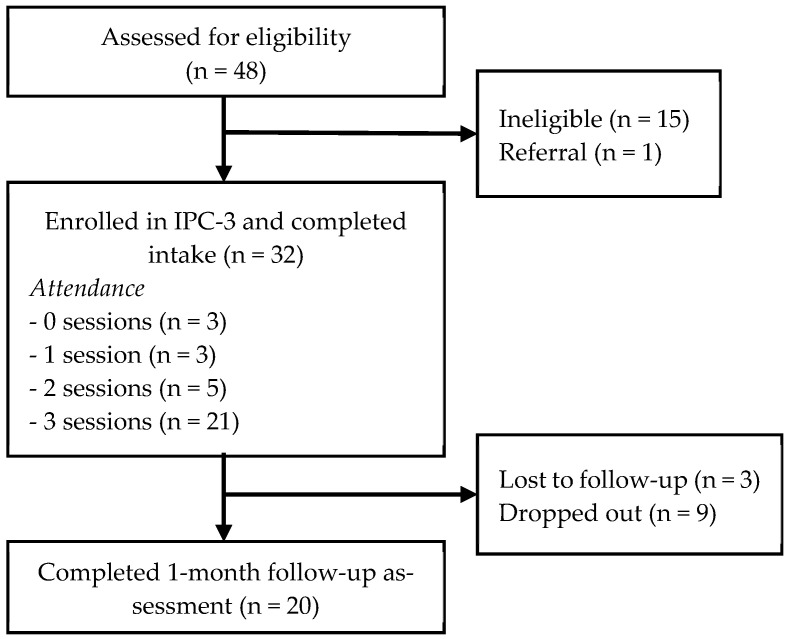
Participant flow diagram.

**Figure 3 ijerph-21-00166-f003:**
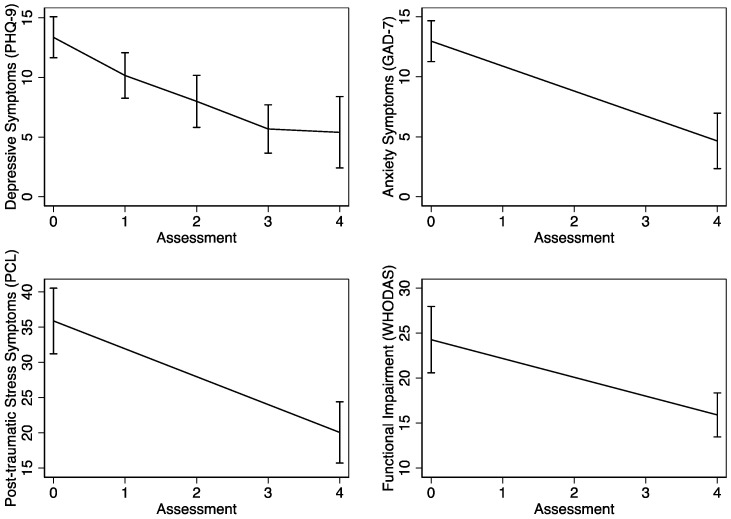
Change in mental health outcomes over time.

**Table 1 ijerph-21-00166-t001:** Data sources and outcomes.

Outcome	Definition [53]	Information Sources	Example Qualitative Questions
Reach	Number, proportion, and representativeness of individuals who participate in IPC-3, including reasons for non-participation.	Key informant interviews (PRE, QUAL)	How would a program like IPC-3 be received by Venezuelan migrants in the community?
Effectiveness	Impact of IPC-3 on mental health and secondary outcomes	Client assessments (PRE and POST, QUANT), Client and provider interviews (POST, QUAL)	Do you think IPC-3 is useful and needed to help people with psychological needs in your community? Why or why not?
Adoption	The number, proportion, and representatives of providers who initiate and deliver IPC-3.	Training records, Providers interviews (PRE and POST, QUAL)	Who do you think is best suited to deliver and receive IPC-3?
Implementation	Fidelity and adaptations made to the components and general implementation of IPC-3.	Key informant, provider, and client interviews (PRE and POST, QUAL), FRAME adaptation tool	Please describe how [IPC-3 component] was implemented? Did you have to change any aspects of IPC-3 in order to make it work? Under what conditions should IPC-3 be delivered remotely (vs. in-person)?
Maintenance	The sustained impacts and implementation of IPC-3	Key informant, provider, and client interviews (PRE and POST, QUAL)	Do you think it is important to try to continue delivering IPC-3 to more people in your community in the long term? What would need to be in place in order to deliver it in the long term?

Abbreviations: PRE: Pre-implementation, POST: Post-implementation, QUAL: Qualitative, QUANT: Quantitative.

**Table 2 ijerph-21-00166-t002:** Characteristics of the study sample.

	Key Informants (n = 10)	IPC-3 Providers (n = 9)		IPC-3 Clients (n = 32)
Age (in years), M (SD)	37.8 (6.1; Range: 30–48)	36.4 (7.6; Range: 23–50)		36.5 (12.2; Range: 23–71)
Female gender, n (%)	10 (100.0%)	9 (100.0%)		31 (96.9%)
Venezuelan, n (%)	7 (70.0%)	9 (100.0%)		32 (100.0%)
Time in Peru (in years), M (SD)	4.71 (0.7)	2.9 (0.8)		2.9 (1.4)
Role and employment, n (%)				
Community leader/volunteer	1 (10.0%)	2 (22.2%)	Unemployed	14 (43.8%)
NGO or CBO worker	7 (70.0%)	3 (33.3%)	Informal employment	12 (37.5%)
UN representative	2 (20.0%)	--	Part-time employment	4 (12.5%)
NGO worker and community leader	--	4 (44.4%)	Full-time employment	2 (6.2%)
Education				
Primary school	--	--		2 (6.2%)
Secondary school	2 (20.0%)	4 (44.4%)		24 (75.0%)
Advanced degree	8 (80.0%)	5 (55.6%)		6 (18.8%)
Previous MHPSS utilization	9 (90.0%)	7 (77.8%)		6 (18.8%)

**Table 3 ijerph-21-00166-t003:** Overview of adaptations to IPC-3 and its implementation.

Description of Adaptation (What)	When	Pro-/Reactive	Who	Fidelity Consistent	Goal/Reason	Contextual Factors
Selected Venezuelan migrants and refugees who were involved in community programs to be trained as providers	PRE	Proactive	HIAS staff, IPC trainers, Researchers	Yes	Increase reach, engagement, acceptability, sustainability, and fit; Address cultural factors	Cultural norms; Competencies
Delivered IPC-3 remotely (online/phone) using phones provided by the project to overcome transportation barriers	PRE	Proactive	HIAS staff, IPC trainers, Researchers	Unknown	Increase reach, engagement, retention, and feasibility; Comply with social distancing guidelines	COVID-19; Location accessibility; Available resources
Separated screening process from IPC sessions. HIAS psychologists referred clients to IPC-3 providers who completed the intake, IPC-3 sessions, and follow-up assessments	PRE	Proactive	HIAS staff, IPC trainers, Researchers	Yes	Improve organizational fit and adherence to policies and procedures; Increase reach, engagement, and acceptability	Service structure; Available resources; Perceptions of the intervention; Cultural norms; Competencies
Modify terminology in IPC-3 manual and materials to maintain conceptual equivalence and contextual relevance: ‘malestar emocional’, ‘duelo migratorio’	TRAIN	Proactive	IPC trainers, IPC providers	Yes	Improve fit, Address cultural factors	Cultural and context; First/spoken language
Provide additional supervision and training to manage difficult situations (e.g., suicidality, other risk/safety concerns), confidentiality, and provider–client boundaries	TRAIN	Reactive	IPC trainers, IPC providers	Yes	Improve provider acceptability, satisfaction, fidelity, competencies, and adoption	Previous training and skills; Cultural norms; Competencies; Provider preferences and expectations; Provider clinical judgment
Provided additional compensation to providers for additional training, supervision, and assessment requirements	TRAIN	Reactive	HIAS staff, IPC trainers, Researchers	Yes	Improve acceptability, feasibility, and adoption	Service structure; Available resources; Provider responsibilities
Modify how concepts are presented in IPC sessions to align with norms, particularly concepts related to the recovery role. For example, ‘taking a break’ was reframed as fortifying yourself and your environment. Other examples include ‘duelo migratorio’ (migratory grief), managing disputes in the context of exploitation, migration-related guilt and pressure to provide remittances, and making decisions about migration plans.	TRAIN/ IMP	Reactive	IPC trainers, IPC providers	Yes	Improve fit; Address cultural factors; Improve fidelity and provider competency	Culture and context
Trainers/supervisors incorporated efforts to monitor provider burden and modeling how everyone struggles to normalize provider challenges and their identifying with client experiences	IMP	Reactive	IPC trainers	Yes	Improve provider acceptability and satisfaction	Provider preferences, expectations, and motivation

Abbreviations: PRE = Pre-implementation; TRAIN = Training; IMP = During implementation; POST = Post-implementation.

**Table 4 ijerph-21-00166-t004:** Summary of client outcomes.

Time Point or Test Statistic	Depressive Symptoms (PHQ-9)	Anxiety Symptoms (GAD-7)	Post-Traumatic Stress Symptoms (PCL-C)	Functional Impairment (WHODAS)
Baseline (n = 32), M (SD)	13.4 (4.8)	13.0 (4.7)	35.9 (12.9)	24.3 (9.9)
Endline (n = 20), M (SD)	5.4 (6.4)	4.7 (4.9)	20.1 (9.3)	15.9 (5.2)
ICC	0.517	0.201	0.399	0.523
Effect size, d	1.1	1.4	1.0	0.8
Test statistic, z (p)	3.5 (<0.001)	3.9 (<0.001)	3.3 (<0.001)	3.1 (0.002)
Qualitative themes from post-implementation interviews related to perceived effectiveness				
Provider interviews	Reductions in depressive symptoms, improved mood	Ability to confront challenging situations that used to cause anxiety and feeling overwhelmed	--	Improvements in functioning, physical health, and self-care
Client interviews	Improved mood and self-confidence	Felt calmer and better able to manage situations in their daily life	More capable to process what had happened in their past	Improvements in healthy behaviors (sleep, nutrition) and social functioning
Analytical integration of qualitative and quantitative findings	Both the qualitative and quantitative data revealed reductions in common symptoms of depression	The quantitative data revealed a reduction in anxiety. Qualitative data described this reduction specifically in reference to overwhelming situations and providing the skills to manage those situations	The quantitative data revealed a reduction in symptoms of post-traumatic stress disorder. Trauma-related stress was not referenced by providers during the interviews.	Both the qualitative and quantitative data revealed improvements in functioning, specifically related to self-care, physical health and wellbeing, and social functioning

## Data Availability

The data presented in this study are available on request from the corresponding author. The data are not publicly available due to the conditions of ethical approval.

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
