# Peer review of "Task Sharing and Remote Delivery of Brief Interpersonal Counseling for Venezuelan Migrants and Refugees Living in Peru during the COVID-19 Pandemic: A Mixed-Methods Pilot Study"

_ijerph, 2024, doi:10.3390/ijerph21020166_

Round 1

Reviewer 1 Report

Comments and Suggestions for Authors

This article looks at the use of interpersonal counseling in Venezuelan refugee communities in Peru.  The authors provide an excellent review of the academic literature and identify the gap for their research to fit.  They provide an excellent review of their project in Peru.  Their conclusions show the value of interpersonal counseling in the Venezuelan refugee community.  One interesting result of the study is the use of members of the refugee community as administrators of interpersonal counseling.  I think the authors could explain more of the benefits and challenges of using members of the community as facilitators of counseling in the region.  Also, how would this example help in other refugee communities in other regions?  By including more generalized conclusions, the authors could increase interest in their research and increase readership for the article. 

Thank you for the opportunity to review the article.  Best of luck in future research.   

Reviewer 2 Report

Comments and Suggestions for Authors

The different concepts under investigation should be better substantiated and detailed in the introduction, supporting them with more solid and varied references.

The procedure seems to me to be very well developed and explained, although the sample seems to me to be small and not probabilistic, so the results cannot be generalized.

The discussion is underdeveloped compared to the results section.  It would be interesting for the easy reading and comprehension of the article to resize the exposed sections making a clear and precise description of what is to be studied, the results obtained, a good discussion in the light of what was obtained.

I believe that the nature of the work should justify a section dedicated to the limitations of the work, especially from the methodological point of view.

Best regards.

Reviewer 3 Report

Comments and Suggestions for Authors

The topic the authors have dealt with is certainly important, given the growth in the number of migrants and the support needs they have. The authors applied Interpersonal counseling (IPC-3), a brief session of psychotherapy that can be used by non-specialized people. The phases of the study method are explained in an understandable way and seem correct. The results of the interviews are presented in an orderly manner. The change in symptoms over time is analyzed correctly.

Limitations of the study are indicated, but not adequately discussed; It would be useful to give more attention to this section.

Round 2

Reviewer 2 Report

Comments and Suggestions for Authors

I think the authors have responded appropriately to the suggestions. The article meets the minimum requirements for publication.

A cordial greeting.